# Assessment of the Antigenotoxic Effects of Alginate and ZnO/Alginate–Nanocomposites Extracted from Brown Alga *Fucus vesiculosus* in Mice

**DOI:** 10.3390/polym13213839

**Published:** 2021-11-06

**Authors:** Ragaa A. Hamouda, Asmaa S. Salman, Asrar A. Alharbi, Reem Hasaballah Alhasani, Maha M. Elshamy

**Affiliations:** 1Department of Biology, College of Sciences and Arts Khulais, University of Jeddah, Jeddah 21959, Saudi Arabia; drasmaasalman@gmail.com (A.S.S.); soso1431h@gmail.com (A.A.A.); 2Microbial Biotechnology Department, Genetic Engineering and Biotechnology Research Institute (GEBRI), University of Sadat City, Sadat City 32897, Egypt; 3Genetic and Cytology Department, National Research Center, Cairo 12622, Egypt; 4Department of Biology, Faculty of Applied Science, Umm Al-Qura University, Makkah 21961, Saudi Arabia; rhhasani@uqu.edu.sa; 5Botany Department, Faculty of Science, Mansoura University, Mansoura 35516, Egypt; mahammy@mans.edu.eg

**Keywords:** ZnO-Alg/NCMs, alginate, *Fucus vesiculosus alga*, genotoxicity, MMC

## Abstract

Mitomycin C (MMC) is an alkylating chemotherapy drug that could induce DNA damage and genetic alteration. It has been used as a model mutagen for in vivo and in vitro studies. The current study aimed to evaluate the protective role of Zinc oxide alginate–nanocomposites (ZnO-Alg/NCMs) against MMC–induced genotoxicity in mice. Animals were treated as follows: the control group, the groups treated with Algin (400 mg/kg b.w), the groups treated with ZnO-Alg/NCMs (400 mg/kg b.w), the group treated with MMC, and the groups treated with MMC plus Algin or ZnO-Alg/NCMs. Pre-treatment with Algin and ZnO-Alg/NCMs was repeated for one or seven days. Zinc oxide alginate-nanocomposites (ZnO-Alg/NCMs) were synthesized with the aim of incorporating the intrinsic properties of their constituents as an antigenotoxic substance. In this study, alginate was extracted from the brown marine alga *Fucus vesiculosus*, Zinc oxide nanoparticles were synthesized by using water extract of the same alga, and loaded in alginate to synthesize ZnO-Alg/NCMs. ZnO-NPs and ZnO-Alg/NCMs were characterized by TEM, SEM, EDX, and Zeta potential. The obtained results confirmed that by TEM and SEM, ZnO-NPs are rod shaped which modified, when loaded in alginate matrix, into spherical shape. The physical stability of ZnO-Alg/NCMs was reported to be higher than ZnO-NPs due to the presence of more negative charges on ZnO-Alg/NCMs. The EDX analysis indicated that the amount of zinc was higher in ZnO-NPs than ZnO-Alg/NCMs. The in vivo results showed that treatment with MMC induced genotoxic disturbances. The combined treatment with Algin and ZnO-Alg/NCMs succeeded in inducing significant protection against MMC. It could be concluded that ZnO-Algin/NCMs is a promising candidate to protect against MMC–induced genotoxicity.

## 1. Introduction

Exposure to various endogenous and environmental agents such as metals, pesticides, and alkylating agents, along with therapeutic compounds including antitumor and antibiotics [1], could induce DNA damage, genetic alteration, endothelial dysfunction, and tissue injury. Mitomycin (MMC) is an alkylating DNA reactive antibiotic agent with anti-proliferative properties isolated from the Gram-positive actinobacteria *Streptomyces caespitosus* [2]. It has been used in the treatment of gastric, bladder, pancreatic, and colon cancer [3]. MMC was observed to induce genotoxic stress in human primary endothelial cells [4]. Due to its stable cytogenetic activity, MMC is used as a model mutagen for in vivo and in vitro studies.

Functional foods greatly promise to improve health and prevent chronic diseases [5]. Alginate is an anion polysaccharide with respect to the linear (unbranched) non-repeating copolymers, containing of huge amount of the two glycan monomers, β–D–mannuronic acid, and its C5–epimer α–L–guluronic acid [6]. Alginates are widely used technologically due to their physical characteristics, for example, their stabilizing, thickening, and emulsifying assets, and also due to individual properties, such as gel strength, porosity, or biocompatibility. Their use is increasing in applications like biomaterials for tissue engineering and bio-printing [7]. Alginates have been revealed to be an extremely biodegradable, more available, and lower price alternative to natural biopolymers [8]. Alginates show similarities to pectin in continental plants [9]. Alginates are present in the cell walls of Phaeophyceae, in a crystalline array analogous to the cellulose microfibrils, and also originate in the intercellular matrix in the Phaeophyceae [10]. Alginate oligosaccharide was observed as a nontoxic and biodegradable polymer, and has a bright possibility for biomedical applications [11]. The antioxidant property of alginate oligosaccharide has received significant attention [12]. The nature of oligosaccharides offers further beneficial characteristics such as anti–inflammatory activity [13] and bacteriostatic function [14]. The most unique feature is its antineoplastic activity [15]. Alginate–derived oligosaccharides were observed to display strong and dose-dependent antitumor effects versus Sarcoma 180 [16]. Alginate oligosaccharides suppressed the tumorigenicity of prostate cancer cells [17]. Seven common genera of marine algae belonging to Phaeophyceae are present in the Red Sea shore of Saudi Arabia. These algae *are Hormophysa, Macrocystis, Fucus, Sargassum, Macrocystis, Padina*, and *Turbinaria* [18]. Nanoparticles are growing in relevance due to the numerous applications in many scientific areas. Several nano-metals synthesized by various methods can be used for a diversity of technological applications [19]. Nanoparticles can be synthesis by applying some plant and algae extracts, as reducing and capping agents, due to the substantial contents of these biomaterials contents [20,21]. Many recent studies investigated that various plant extracts can be used to manufacture metal oxide nanoparticles, including, remarkably, zinc oxide (ZnO-NPs). ZnO-nanoparticles are safe and less toxic than other metal oxides [22]. Studies proved that ZnO-NPs have antibacterial, antiviral, and antifungal activity [23]. ZnO-NPs have antibacterial activity, are nontoxic, and economic, and do not show any injurious effects to normal body cells [24]. Nanocomposites are a combination of biopolymers, such as alginate, and inorganic materials, like zinc oxide (ZnO), in nano dimension size, that give mechanical strength, extreme thermal resistance, and low permeability [25,26]. Significant composites of alginate are ionically bonded with different ions such as Ca^+2^, Ag^+1^ [27], and Zn^+2^ [28]. Sodium alginate with ZnO nanocomposite films have a potential application in wound healing [24]. Therefore, this study aimed to synthesize these by using biological algal extract and characterize the generated ZnO/Alg-NCMs by physical means, which subsequently will be investigated for its ability to reduce MMC induced genotoxicity.

## 2. Material and Methods

### 2.1. Alga

*Fucus vesiculosus* and *Padina pavonica* were gathered from the beach at Jeddah, Saudi Arabia and were recognized according to Taylor [29]. The algae were gathered in April 2020. Marine algae were cleaned with running tap water followed by distilled water to completely eliminate any debris. The cleaned algal biomass was then dried in oven at 60 °C until a constant weight was achieved. The dried and grounded algal powders were sieved using the standard laboratory test sieve; the size of algae particles were 1–1.2 mm.

### 2.2. Alginate Extraction

The Na-alginate extraction method was performed according to the method developed by Hambali et al. [30]. In total, 100 gm of *F. vesiculosus* and *P*. *pavonica* were soaked with 1% HCl for 1 h. Then, it was rinsed with 3% Na_2_CO_3_ solution in relation to 1:30 (*w/v*) at 65 °C for 150 min, and then filtration was performed. Then, 10% HCl solution was amended in relation of 1:1 and allowed to form into white alginic acid to a pH of 2–3 within 30 min. Extreme pH was neutralized by the addition of dilute Na_2_CO_3_, with blending to ensure homogenization of the solution, until a pH of 7 was attained. After a pH of 7 was attained, Isopropyl alcohol (IPA) was added to obtain the Na alginate, and then dried at 60 °C in an electrical oven.

### 2.3. Alginate Yield

The Na-alginate yields that were obtained from the *F. vesiculosus* and *P. pavonica* extraction process were calculated, based on the weight of Na-alginate after drying, compared to the algal dry weight. The calculation of Na-alginate yield percentage was achieved using the following formula:(1)$$\mathrm{Yields}\;\%=\frac{A}{D}*100$$

D is the initial dry weight of algae used, A is the weight of alginate.

### 2.4. Alginates Characterisation

Fourier transforms infrared Spectroscopy (FTIR) is an important device that is used to identify the functional groups that relate to alginate. The previously extracted dry alginate was analyzed using FTIR spectroscopy and compared with standard alginic acid obtained from Sigma (Saudi Investment Group and Marketing) Ltd. Dry alginate samples were mixed with pellets of potassium bromide and the FTIR spectra were then investigated within the range of 400–4000 cm^−1^ using a “Termo Fisher Nicolete IS10, (Waltham, MA, USA) Spectrophotometer”. ^1^H NMR spectra were acquired in 0.1% *w/v* solutions of sodium alginate in DEMSO. Negative and positive-ion electrospray mass spectrometry (ESI-MS) was carried out on a Micromass Q-TOf and Q-TOf Ultima instruments (Waters, Manchester, UK). Nitrogen was used as the desolvation and nebulizer gas at a flow rate of 250 and 15 L/h, respectively. Source temperature was 80 °C and the desolvation temperature was 150 °C.

### 2.5. Green Synthesis of Zinc Oxide Nanoparticles

Zinc oxide nanoparticles were synthesized by using *F. vesiculosus*. Zinc acetate dehydrate (0.02 M) was supplemented to 40 mL of D water under constant stirring. Ten milliliters of algal aqueous extract (1 gm of alga added to 100 mL D water and boiled for 1 h) was added dropwise to this solution with stirring. After 10 min, NaOH (2.0 M) was added until reaching pH 12 and a pale white aqueous solution was obtained. The precipitate was filtered and washed twice with DD water followed by ethanol to obtain a solution free of contaminations. The ZnO-NPs were yielded after drying at 60 °C in a vacuum oven overnight [31,32].

### 2.6. Synthesis of ZnO/Alginates Nanocomposites (ZnO/Alg-NCMs)

A total of 2 gm of Sodium alginate were dissolved in 100 mL D water, at 40 °C, then 0.075 g ZnO-NPs was added, and the mixture was stirred for 2 h. After 2 h the solution was filtrated and dried at 60 °C for 2 days.

### 2.7. Characterizations of Zinc Oxide Nanoparticles and ZnO/Alg-NCMs

The characterization of ZnO/Alg-NCMs was performed by Fourier transform infrared spectroscopy (FTIR) using a (JASCO Europe S.r.l., Cremella, Italy), FT-IR 5300 spectrophotometer. Morphology and particle size of the nanoparticles were characterized via scanning electron microscopy JEOL JSM-6510/v, Tokyo, Japan (SEM), and transmission electron microscopy (JEOL JSM-6510/v, Tokyo, Japan), (TEM). The chemical contents of the nanoparticles were scanned by energy dispersive X-ray spectroscopy (EDS) using a Zeta potential analyser (Malvern Zeta size Nano-Zs90) Malvern United States.

### 2.8. Chromosome Abnormalities in Bone Marrow Cells

An investigation of chromosomal aberration was conducted in the bone marrow of mice injected intraperitoneally with colchicine, two hours before euthanasia [33]. Chromosome extractions from bone marrow cells were taken according to the method of Yosida and Amano [34]. One hundred well spread metaphases were analyzed per mouse and the metaphases with gaps, chromosome or chromatid breakage, and fragments were documented. Metaphases with gaps, chromosome or chromatid breakage, and fragments were recorded.

### 2.9. Chromosome Abnormalities in Spermatocytes

The diakinase-metaphase I cells collected from the spermatocytes were made following the air-drying technique of Evans et al. [35]. Slides were stained with 7% Giemsa stain in phosphate buffer (pH 6.8). A total of 500 well spread metaphases (5 animals/group) were scored for chromosome aberrations in germ cells. The types of aberrations in spermatocytes including XY univalent, autosomal univalent, and fragments were scored.

### 2.10. Sperm Morphology Assay

Sperm morphology assays were performed according to Rasgele [36]. Both of the cauda epididymides from each animal were dissected, excised, and minced in isotonic sodium citrate solution (2.2%). Smears were prepared and sperms were stained with Eosin Y [36]. At least 1000 sperms per animal (5 animals/group) were assessed for morphological abnormalities of the sperm shape. Sperm head and tail abnormalities were determined as having either normal or abnormal morphology.

### 2.11. Statistical Analysis

The genotoxicity analysis, the significance of each treatment, was assessed by *t*-test. All statements of significance were established on a probability of *p* ≤ 0.05.

## 3. Results and Discussion

### 3.1. Alginate Yields

Results in Figure 1 summarize that *F. vesiculosus* denoted larger amounts of alginate yields than *P. pavonica*. Both algae were collected from the beach at Jeddah, in the Red Sea in the same season. Alginate yields and quantity depended on various influences, for instance season of their collecting, type of algal age and brown algal species [37,38,39], age, harvesting methods [40], and extraction methods [41]. The alginates extracted from *F. vesiculosus* are more susceptible to high temperatures than alginates from any other species [10].

### 3.2. Fourier-Transform Infrared Spectroscopy (FTIR) Analysis

FTIR spectroscopy of the alginic acid (slandered) and FTIR analysis of alginate extracted from brown alga *F. vesiculosus* were investigated, Figure 2 and Table 1. There are 17 bands regions 3448, 2923, 2855, 1620, 1418, 1308, 1620, 1418, 1308, 1126, 1095, 1031, 947, 892, 819, 781, 725, 675, 622, and 433 cm^−1^. However, there are 15 bands in extracted alginate from *F. vesiculosus*, the bands are 3449, 2992, 2854, 1653, 1625, 1519, 1465, 1402, 1273, 1115, 1048, 870, 672, 588, and 528 cm^−1^. The broad peak present at 3448 in alginic acid is shifted to 3449 cm^−1^ in alginate extracted from *F. vesiculosus*, and very narrow peaks at 2923 and 2855 present in alginic acid are shifted to 2922 and 2854 cm^−^1, respectively, in alginate extracted from *F. vesiculosus*. The peak present at 1620 cm^−1^ in alginic acid shifted to 1625 cm^−1^ in alginate extracted from *F. vesiculosus*, however two others new peaks, 1653 and 1519 cm^−1^, are present in the same regions in alginate extracted from *F. vesiculosus*. The wave number 3448 cm^−1^ indicates OH bonds in alginic acid [42], the peak present in wave number 2923 cm^−1^ denotes C–H stretching bands [43]. The wave number (2855) reveals C-H functional group [44]. The wave number 1620 cm^−1^ relates to the peak of carbonyl stretching [45]. The wave number 1418 cm^−1^ indicates deformation of C–H [42]. The band found at wave number 1126 cm^−1^ relates to *n*(C–O) [46]. The wave number present at 1095 cm^−1^ indicates PO2 symmetric (Phosphate II) [47]. The wave number 1031 cm^−1^ indicates C-O stretching [48]. The bands present in wave number 892 cm^−1^ relates to C–C and C–O [42]. The bands found at wave number 781 cm^−1^ relate to Out-of-plane bending vibrations [49], the wave number 725 cm^−1^ relates to Out-of-plane bending vibrations [49], and the peak present at wave number 675 cm^−1^ relates to CH out-of-plane bending vibrations [50].

### 3.3. Electrospray Mass Spectrometer

Alginate standard was used in ESI-MS analysis which indicated the great similarity to sample that isolated from brown alga. Negative and positive-ion ESI-MS/MS was initially evaluated with various pseudo-molecular ions, including [M − H], [M + H], [M + Na], and [M − Cl], as precursors for optimal sequence information (Figure 3).

### 3.4. Proton Nuclear Magnetic Resonance (H^1^ NMR) Analysis

H^1^ NMR spectroscopy is an important device for determining the structure of polysaccharides. Structural features of alginates are represented by H^1^ NMR Figure 4. Results denote the relation of β-anomeric protons in the alginate sample in addition to the entrance of protons signals within 4.5 to 5.7 ppm. Signals recognized to the anomeric hydrogen of guluronic acid (G) at 5.66–5.70 pm and the H-5 of mannuronic acid at 4.56–5.03. These results are agreement with [56]. The ^1^H NMR profile exposed the incidence of guluronic acid H-5 (GG-5G) that was determined at 4.282 ppm (signal III), as reported by Usoltseva et al. [57].

### 3.5. SEM and TEM Image

The shape and distribution of biosynthesis ZnO-NPs and ZnO/Alg-NCMs that were biosynthesized by brown alga *F. vesiculosus*, and alginates extracted from the same alga investigated by Scanning Electron Microscope (SEM). The rod shape was observed with ZnO-NPs and ZnO/Alg-NCMs Figure 5. The surface of ZnO/Alg-NCMs is observed on fibrous nanocomposite surfaces from sodium alginate and ZnO-NPs indicating that ZnO nano has been effectively modified on the surface of sodium alginate [58]. The results proved by the TEM image in Figure 5 demonstrated that ZnO-NPs are rod shaped, one diameter size ranges from 20 to 38 nm, meanwhile ZnO/Alg-NCMs are a spherical shape with size ranges from 7.75 to 17.75 and a high distribution. The results shown in Figure 5 demonstrate that ZnO-NPs was modified when embedded with alginate due to ZnO-NPs being rod shaped and ZnO/Alg-NCMs being a spherical shape. Trandafilović et al. [59] demonstrated micrographs of ZnO-dispersed NPS showing a huge number of spherical particles well spread in the alginate matrix. Nano-ZnO in the sodium alginate is well distributed [60]. Hamouda et al. [31,32] observed the green-synthesized ZnO-NPs by green alga *Ulva fasciata* had a rod shape.

### 3.6. Zeta Potential

The constancy of the attained nanomaterials was investigated by the zeta potential measurement [61]. Figure 6 reported the Zeta potential of the apparent surface charge of zinc oxide nanoparticles, and ZnO/Alg-NCMs bio-fabricated by brown alga *F. vesiculosus*. The results show the Zeta potential value was −1.91 and −3.78 of biosynthesis zinc oxide nanoparticles and ZnO/Alg-NCMs, respectively. The negative charge indicated that the highest physical stability of ZnO/Alg-NCMs was more than ZnO-NPs bio–synthesised by *F. vesiculosus*. Haider and Mehdi [62] reported the value of the negative charge of the zeta potential shows the efficacy of the capping compounds in stabilizing AgNPs by denoting more negative charges that reserve all the particles away from each other. The negative values explain the repulsion between the particles and, thereby, realization of more stability of the AgNPs structure eluding agglomeration in aqueous solutions [63]. The zeta potential value of nanocomposites is greater than the zeta potential of ZnO NPs synthesized intracellular with *Lactobacillus plantarum* [61].

### 3.7. Energy Dispersive X-ray Measurements

The results in Figure 7 represent peaks of metals content of the zinc oxide nanoparticles and ZnO/Alg-NCMs bio-fabricated by the brown alga *F. vesiculosus* that were assessed by Energy dispersive X ray spectrophotometry. The five metals are C, O, Ca, Cu, and Zn with weight percentage 31.12, 28.34, 0.33, 2.22, and 37.98, respectively. Meanwhile seven metals are presented in case of ZnO/Alg-NCMs bio-fabricated by brown alga *F. vesiculosus*: C, O, Na, Ca, Cu, Zn, and Sb with weight percentage 28.78, 41.61, 4.44, 13.46, 2.78, 3.28, and 5.65, respectively. The results suggested that some metals, such as Ca and Na, present in alginates, were not present in ZnO-NPS, and also the weight percentage of carbon and oxygen are higher in ZnO/Alg-NCMs than those present in ZnO-NPS, due to the fact that carbon and oxygen are the main contents of alginate polymers. The existence of a sodium (Na) peak proves the presence of sodium alginate [64].

### 3.8. Chromosomal Aberrations in Bone Marrow Cells

Table 2 shows the number and percentage of the chromosomal aberrations induced in control and treated animals. The mean percentage of metaphases with chromosomal aberrations reached 24.6% (*p* < 0.01) 24 h after intraperitoneal injection with Mitomycin C. The percentages of aberrant cells in animals treated with alginate (Algin) and ZnO/Alg-NCMs were statistically not significant compared to the control group. Treatment with alginate and ZnO/Alg-NCMs for 7 days caused a highly significant (*p* < 0.01) reduction in the percentage of chromosomal abnormalities induced by MMC. The percentage of reduction reached 33.3 and 56.9% after treatment with alginate and ZnO/Alg-NCMs, respectively, Table 2. Additionally, Table 2 illustrates the protective effect of alginate and ZnO/Alg-NCMs in reducing the different types of aberrations, Figure 8 demonstrates different types of chromosomal aberrations observed in bone marrow cells.

### 3.9. Chromosome Abnormalities in Spermatocytes

A highly significant (*p* < 0.01) elevation in spermatocytes’ chromosomal aberration was observed after MMC treatment which reached 17.6 ± 0.44%. Treatment with Alginate and ZnO/Alg-NCMs for 7 days ameliorated these genotoxicities and reduced the aberrations statistically in a highly significant (*p* < 0.001) manner (Table 3). XY-univalents and autosomal univalents were the main aberrations observed in diakinase metaphase I cells Figure 9.

### 3.10. Sperm Shape Abnormalities

As shown in Table 4, MMC induced a statistically highly significant (*p* < 0.01) percentage of sperm abnormalities in male mice which reached 14.2%. The dominant abnormalities found were amorphous, triangular, and without hook heads or coiled tail. Simultaneous treatment of mice with Alginate and ZnO/Alg-NCMs reduced the percentage of sperm abnormalities. It reached 8.64 and 7.16% (*p* < 0.01), respectively. The percentage of reduction reached 39.15 and 49.57% after treatment of mice with Alginate and ZnO/Alg-NCMs, respectively, Figure 10.

Chromosome abnormalities in bone marrow cells were employed as a cytogenetic end point in genetic risk assessment. One of the most significant bioassays for assessing the genotoxicity of various substances is in vivo chromosomal abnormalities [65,66]. In the present study, MMC induced a high and statistically significant percentage of chromosome aberrations in mouse bone marrow cells. These finding are in agreement with those observed by Khalil et al. [67] who documented that MMC induced chromosomal aberrations in Swiss albino mice. MMC has been shown to cause mitotic delay [68] which could affect the yield of chemically induced abnormalities [69]. MMC is an alkylating agent that induces DNA damage leading to an arrest of replication and transcription as well as apoptosis [70]. The activated metabolite of MMC mitosene reacts via *N*-alkylation with 7-*N*-guanine nucleotide residues in DNA at the location of 5′–CpG–3′ sequence that causes DNA crosslinking and, thus, inhibits DNA synthesis [71].

MMC induced a high and statistically significant percentage of chromosome abnormalities in diakinesis metaphase I mouse spermatocytes. The most common type of abnormality seen in mouse spermatocytes after treatment with MMC was the univalents. There are reports that the X and Y chromosomes occasionally separate to form univalents in mouse primary spermatocytes [72]. It is termed non-pairing of XY, or X-Y dissociation. Leonard and Linden [73] concluded that autosomal univalents resulting from asynapsis or desynapsis are rare in control animals, since chiasma formed at the diploten stage maintain pairing association until the end of metaphase. X and Y chromosomes do not have homologous segments and so, X-Y univalents are the most common aberration. In the first meiotic prophase, homologous chromosomes undergo pairing, chiasma formation and crossing over [74]. In bivalents, when the process of pairing fail to progress in a normal manner, the two homologs would persist as unassociated univalents which could undergo random separation at anaphase I, leading to aneuploidy in the following metaphase.

The present study demonstrated that the mean percentages of sperm shape abnormalities were significantly increased after MMC treatment. These results support those obtained by Kumari et al. [75]. Sperm shape abnormalities reflect changes in the DNA content [37]. Also, sperm head abnormalities are frequently taken as a typical test for monitoring the mutagenic potential for many chemicals [76]. Tail deformities were conveyed to decrease fertility in human and animals [77].

The results indicated that pre-treatment with alginate for 7 days significantly reduced chromosomal aberrations in somatic and germ cells. Alginate oligosaccharide was observed to decrease H_2_O_2_-induced oxidative stress [11]. Guo et al. [13] documented that pre-treatment of mice with alginate oligosaccharide inhibited the oxidative stress process and were well protected against acute Doxorubicin cardiotoxicity. Oligosaccharides identified from brown algae were found to exhibit an anti-proliferative activity via induction of cell cycle arrest and apoptosis on human prostate cancer cells both in vitro and in vivo [17].

The present results showed that pre-treatment with ZnO-Alg/NCMs for 7 days significantly reduced chromosomal aberrations in somatic and spermatocytes and reduced the percentage of morphological sperm abnormalities. Bionanocomposites are a combination of biopolymers and inorganic materials, mainly from metal oxides like silver nitrate, titanium oxide, silicon oxide, and zinc oxide in nano dimension size. The incorporation of zinc oxide nanoparticles into sodium alginate possesses an antibacterial effect which inhibits the growth of various Gram-positive pathogens [25]. Nystrom and Bisrat, [78] showed that such a relationship between the specific surface dissolution rate and the solubility of equilibrium increased with a decrease in particle size. The hyperbolic relationship between particle size and surface-specific dissolution rate increases solubility [79] due to the larger surface area, which promotes dissolution [80]. Moreover, a decline in the particle size into nanoparticles enhances the solubility and bioavailability of the active ingredients [81].

## 4. Conclusions

The current study revealed *F. vesiculosus* denoted more alginate yields than *P. pavonica. FT-IR*, ESI-MS and ^1^H NMR proved that the compound that was extracted from *F. vesiculosus* is alginate. TEM, SEM, EDX, and Zeta potential proved that ZnO/Alg-NCMs is spherical in shape, more negatively charged than ZnO-NPs, and the amount of zinc in ZnO/Alg-NCMs was lower than in the case of ZnO-NPs. The current study also revealed that MMC increased chromosomal abnormalities in bone marrow and spermatocytes and significantly elevated the percentage of sperm shape deformities. Oral administration of ZnO-Alig/NCMs was found to be safe and may be a potential candidate to ameliorate the detrimental effects of different chemical-induced oxidative damage and cytogenetic alterations.

## Figures and Tables

**Figure 1 polymers-13-03839-f001:**
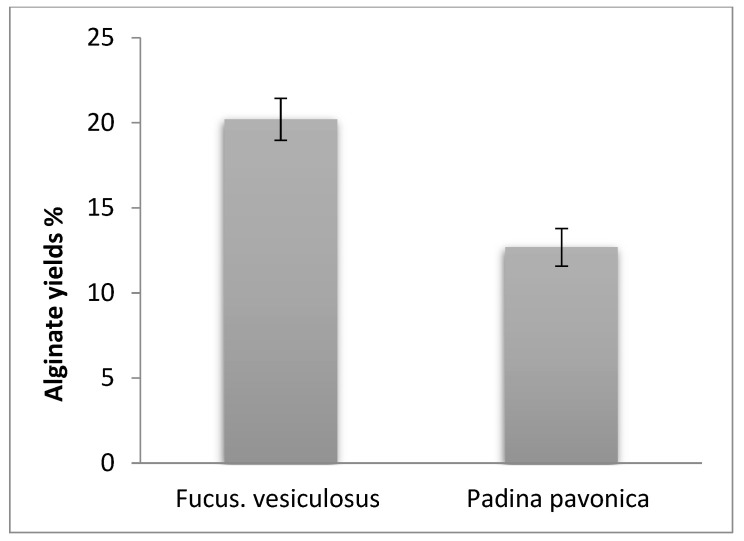
Alginate yield.

**Figure 2 polymers-13-03839-f002:**
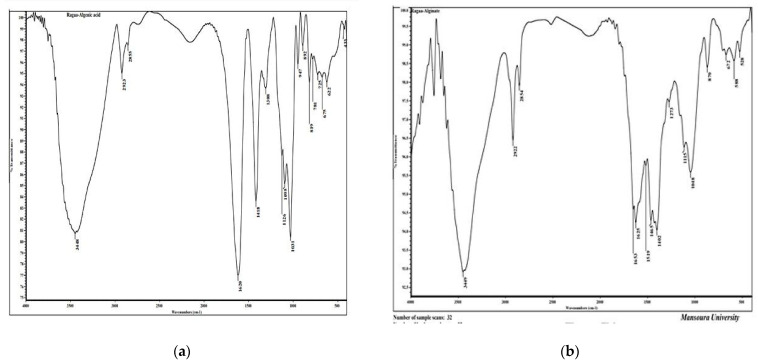
Alginic acid (**a**) and alginate (**b**) analysis by FT-IR.

**Figure 3 polymers-13-03839-f003:**
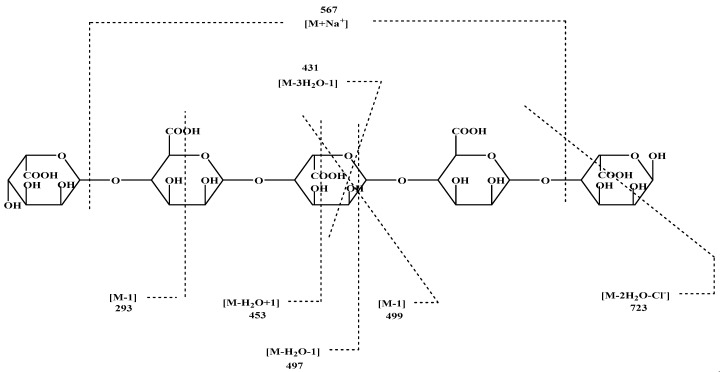
Negative and positive-ion ESI-MS/MS product-ion spectra of Alginate (GMGMG).

**Figure 4 polymers-13-03839-f004:**
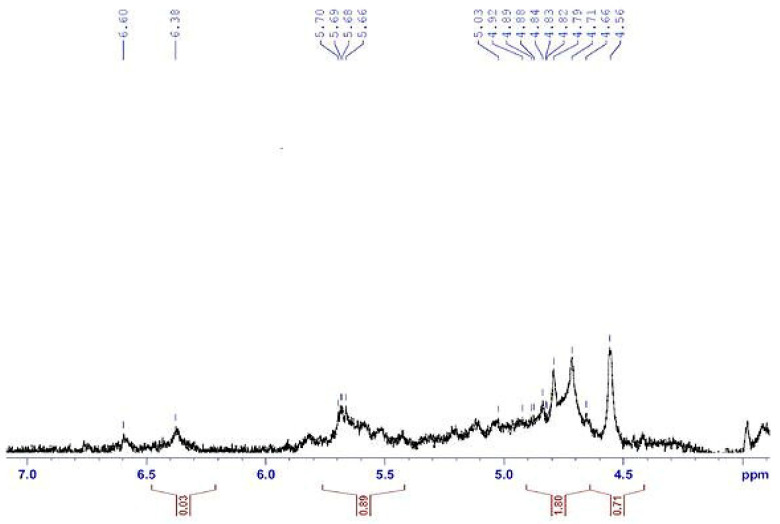
^1^H-NMR spectra (500 MHz) for solutions of sodium alginate extracted from *F. vesiculosus*.

**Figure 5 polymers-13-03839-f005:**
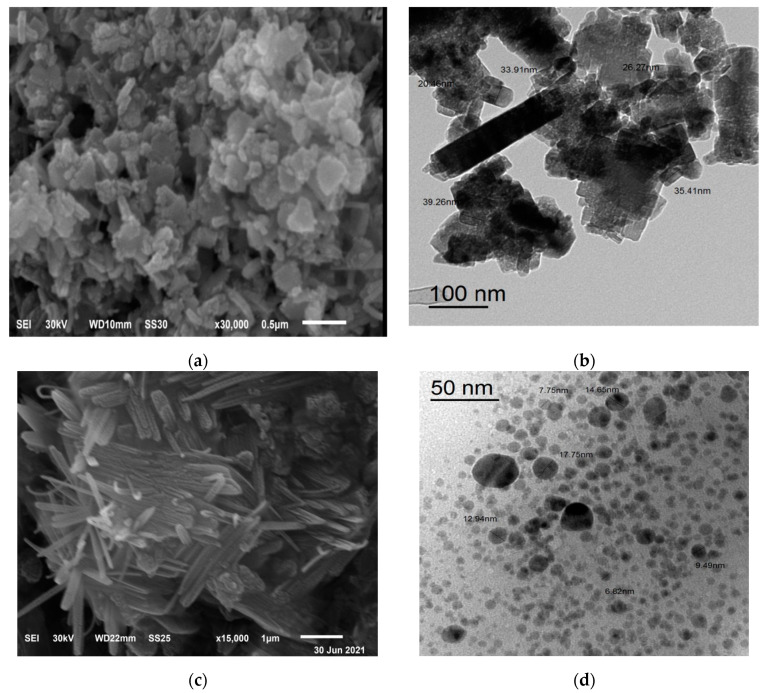
Transmission and scanning Electron Microscopic image, SEM (**a**,**c**) and TEM (**b**,**d**), of Biosynthesis zinc oxide nanoparticles and ZnO/Alg-NCMs.

**Figure 6 polymers-13-03839-f006:**
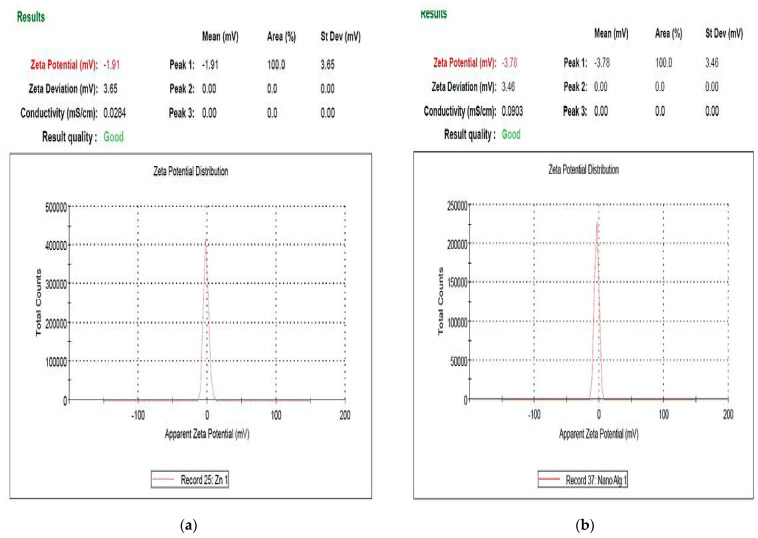
Zeta potential analyses of zinc oxide nanoparticles (**a**) and ZnO/Alg-NCMs (**b**) bio–fabricated by brown alga *F. vesiculosus*.

**Figure 7 polymers-13-03839-f007:**
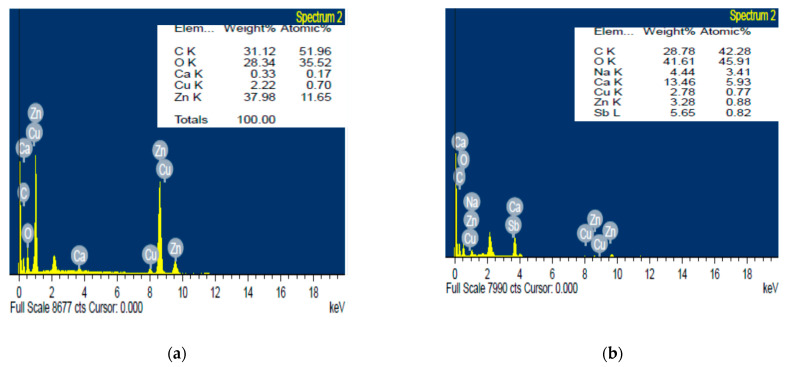
Energy dispersive X-ray spectrophotometry analysis of zinc oxide nanoparticles (**a**) and ZnO/Alg-NCMs (**b**) bio-fabricated by brown alga *F. vesiculosus*.

**Figure 8 polymers-13-03839-f008:**
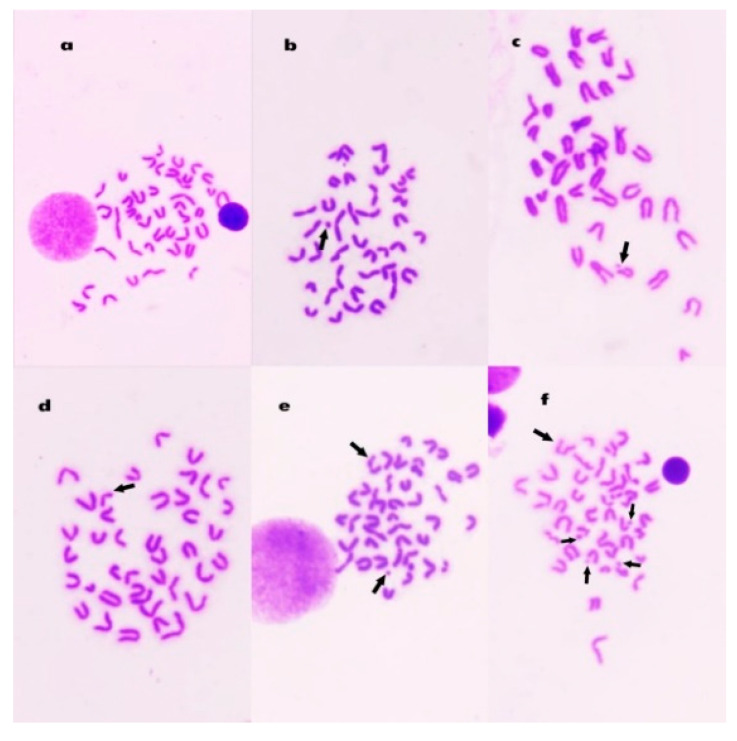
Metaphase plates from mouse bone marrow cells showing (**a**)normal; (**b**) fragment; (**c**) break; (**d**) deletion; (**e**) gap and fragment; and (**f**) break, fragment, and gap.

**Figure 9 polymers-13-03839-f009:**
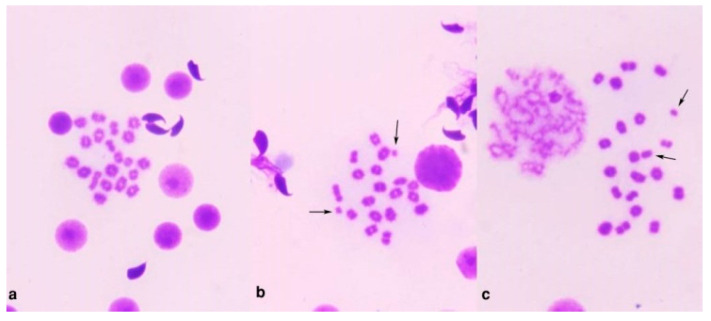
Diakinesis-metaphase I plates of meiosis showing (**a**) normal, (**b**) autosomal univalents; and (**c**) X-Y univalent.

**Figure 10 polymers-13-03839-f010:**
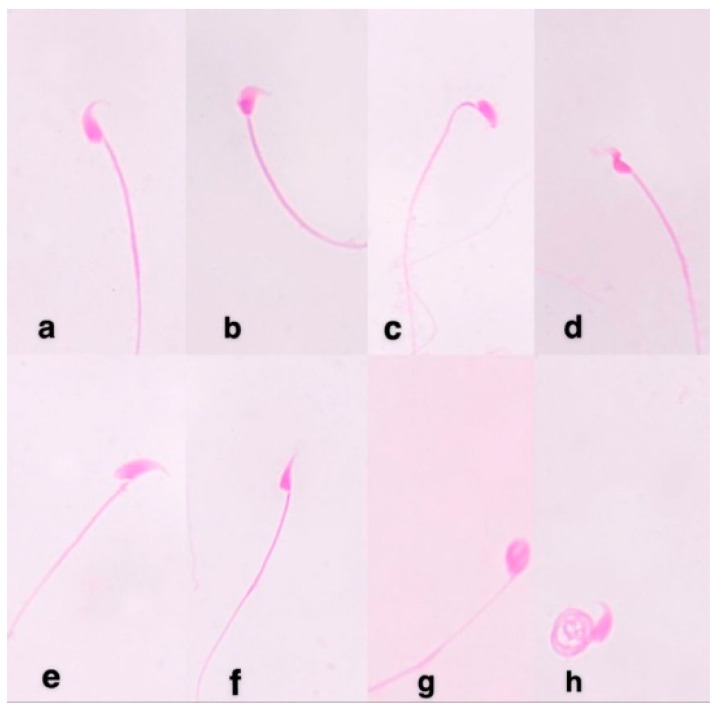
Sperm abnormalities in male mice showing: (**a**) normal; (**b**) triangle; (**c**) without hook; (**d**) banana; (**e**) big; (**f**) small; (**g**) amorphous head; and (**h**) coiled tail.

**Table 1 polymers-13-03839-t001:** Alginic acids and alginate analysis by FT-IR.

Alginic Acids Wavenumber [cm^−1^]	Alginates Wavenumber [cm^−1^]	Assignment	Ref. No.
3448	3449	OH bonds	[42]
2923	2922	Asymmetric stretching vibration of CH_2_ of acyl chains (lipids)	[32]
2855	2854	CH_2_ symmetric stretching	[5]
	1653	C D O, C D N, NH of adenine, thymine, guanine, cytosine	[51]
1620	1625	Amide II	[28]
	1519	Amide II	
	1465	CH_2_ scissoring mode of the acyl chain of lipid	[52]
1418	1402	d C–H, d C–O-H	[53]
1308	1273	CHaα– rocking	[54]
1126	1115	Symmetric stretching P–O–C	[47]
1048	1048	C=O groups	[55]
892–819	870	Vibrations of aromatic ring	
781–725			
675–622	672	800–600 C–Cl	[31,32]
-	588	750–500 C–I	[31,32]
433	528	750–500 C–I	[30,31]

**Table 2 polymers-13-03839-t002:** Number and mean percentage of the different types of chromosomal aberrations in mouse bone marrow cells after treatment with different doses of alginate (Algin) and alginate-nanocomposite (ZnO/Alg-NCMs) alone or in combination with MMC.

Treatment and Doses (mg/kg b.wt.)	Treatment Day(s)		No. of Metaphases with				Chromosomal Aberrations		Inhibition %
		Gap	Frag. and/or Break	Gap+(Frag. or Break)	Deletion	Rt.	Excluding Gaps Mean ± S.E.	Including Gaps	
MMC	1	24	65	26	4	4	19.8 ± 0.26 ^a^	24.6 ± 0.4 ^a^	
Control		10	15	-	-		3.0 ± 0.2	5.0 ± 0.27	
Algin 400		7	14	2	-	-	3.2 ± 0.2	4.6 ± 0.21	
ZnO-ALg/NCMs 400		10	13	1	-	-	2.8 ± 0.21	4.8 ± 0.23	
AL 400 + MMC		18	56	24	4	2	17.2 ± 0.7 ^a^	20.8 ±0.27 ^a^	15.44
ZnO-Alg/NCMs + MMC		14	52	26	1	-	15.8 ± 0.45 ^a^	18.6 ±0.3 ^a^	24.39
MMC	1	24	65	26	4	4	19.8 ± 0.26 ^a^	24.6 ± 0.4 ^a^	
Control	7	12	15	1	-	-	3.2 ± 0.37	5.6 ±0.3	
Algin 400		11	13	-	-	-	2.6 ± 0.3	4.8 ± 0.23	
ZnO-Alg/NCMs 400		9	14	1	-	-	3.0 ± 0.3	4.8 ± 0.3	
AL 400 + MMC		21	48	13	-	-	12.2 ± 0.3 ^ab^	16.4 ± 0.3 ^ab^	33.33
ZnO-Alg/NCMs + MMC		15	24	14	-	-	7.6 ± 0.2 ^ab^	10.6 ± 0.34 ^ab^	56.9

The total number of scored metaphases is 500 (5 animals/group); Frag. = fragment, Rt. = Robertsonian translocation. ^a^: significant at 0.01 level (*t*-test) compared to control (non-treated). ^b^: significant at 0.01 level (*t*-test) compared to treatment.

**Table 3 polymers-13-03839-t003:** Number and mean percentage of metaphases with chromosomal aberrations in mouse spermatocytes after treatment with Alginate and ZnO/Alg-NCMs alone or in combination at different doses of Alginate with MMC.

Treatment and Doses (mg/kg b.wt.)	Treatment Day(s)	No. of Different Types of Chromosomal Aberrations			Total Aberrations		Inhibition %
		XY Univalent	Autosomal Univalent	XY+ Autosomal Univalent	No.	Mean % ± S.E.	
MMC	1	46	27	15	88	17.6 ± 0.44 ^a^	
Control		13	4	-	17	3.4 ± 0.2	
AL 400		14	4	-	18	3.6 ± 0.24	
ZnO-Alg/NCMs 400		15	3	1	19	3.8 ± 0.2	
Alginate 400 + MMC		42	27	10	79	15.8 ± 0.4 ^a^	10.2
ZnO-Alg/NCMs + MMC		43	25	9	77	15.4 ± 0.36 ^a^	12.5
MMC	1	46	27	15	88	17.6 ± 0.44 ^a^	
Control	7	13	6	-	19	3.8 ± 0.24	
Alginate 400		12	4	-	16	3.2 ± 0.2	
ZnO-Alg/NCMs 400		13	5	-	18	3.6 ± 0.22	
Alginate 400 + MMC		28	10	7	45	9.0 ± 0.5 ^ab^	48.8
ZnO-Alg/NCMs + MMC		22	9	5	36	7.2 ± 0.33 ^ab^	59.09

The total number of scored metaphases is 500 (5 animals/group); ^a^: significant at 0.01 level (*t*-test) compared to control (non-treated); ^b^: significant at 0.01 level (*t*-test) compared to treatment.

**Table 4 polymers-13-03839-t004:** Number and percentage of different types of sperm shape abnormalities in male mice after treatment with different doses of Alginate (Algin) and alginate-nanocomposite (ZnO/Alg-NCMs) alone or in combination with MMC.

Treatment and Doses (mg/kg b.wt.)	Examined Sperm No.	No. of Sperms with						Abnormal Sperm No.	Abnormal Sperms Mean % ± S.E.	Inhibition %
		Head Abnormalities					Tail Abnormalities			
		Amorphous	Triangle	Without Hook	Small	Big	Coiled			
MMC	5000	171	163	135	6	5	230	710	14.2 ± 0.21 ^a^	
Control	5000	34	27	19	-	-	15	95	1.9 ± 0.25	
Alginate 400	5000	37	23	14	-	-	10	84	1.68 ± 0.25	
ZnO-Alg/NCMs 400	5000	33	22	19	-	-	17	91	1.82 ± 0.33	
AL 400 + MMC	5000	95	112	72	6	2	145	432	8.64 ± 0.43 ^ab^	39.15
ZnO-Alg/NCMs 400 + MMC	5000	88	109	68	5	1	87	358	7.16 ± 0.7 ^ab^	49.57

^a^: significant at 0.01 level (*t*-test) compared to control (non-treated). ^b^: significant at 0.01 level (*t*-test) compared to treatment.

## Data Availability

The datasets spent and/or analyzed during this study are available from the corresponding author on reasonable request.

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
