# Peer review of "Assessment of the Antigenotoxic Effects of Alginate and ZnO/Alginate–Nanocomposites Extracted from Brown Alga Fucus vesiculosus in Mice"

_polymers, 2021, doi:10.3390/polym13213839_

Round 1

Reviewer 1 Report

The overall idea of this work is interesting and we have seen increasing numbers of research looking into plant-derived substance and plant-assisted/associated synthesis of NPs for the purpose of bacterial resistance, cancer treatment, and other health related applications. This article/work is reasonably organized in terms of scientific content but utterly needs some improvements in the following aspects:

  1. The motivation of this research was not well explain in the introduction section. The author should make efforts to better address the problem and broad impact of MMC (particularly in numbers and facts). Also the logic in introduction has to be reorganized for a more clear story-telling purpose.
  2. There are significant flaws in the experiment setup. ZnO-NPs has to be added as an independent group and with MMC to demonstrate that the antigenotoxic effect is solely from ZnO-Algin/NCMs but not Algin or ZnO-NPs alone. Please revise accordingly.
  3. Formatting issues: the font type and size are not the same in some paragraphs of this article, e.g. section 2.4 and section 3.2; figure and table numbers are not properly cited, eg. figure 4; period and comma characters missing in some places; figure title is not informative, e.g. not all images are clearly explained and identified in Figure 5; reference not formatted correctly;
  4. Grammar issues: many sentences in the article are not properly written. Paragraphs are scattered into pieces of information other than being logically and smoothly laid out. Some grammar problems are preventing the reader to understand the content correctly and effortlessly.

Author Response

Thank you very much for giving us a chance to respond to the reviewers’ comments. We have made every effort to address each reviewer’s concerns as clearly and succinctly as possible, as demonstrated in the following pages.

The changes made in the text as requested from reviewers are highlighted in red colour in the text.

Thank you once again for this opportunity

Sincerely,

Prof: Ragaa Hamouda

Comments and Suggestions for Authors

The overall idea of this work is interesting and we have seen increasing numbers of research looking into plant-derived substance and plant-assisted/associated synthesis of NPs for the purpose of bacterial resistance, cancer treatment, and other health related applications. This article/work is reasonably organized in terms of scientific content but utterly needs some improvements in the following aspects:

Thank you

Reviewers comments

  1. The motivation of this research was not well explain in the introduction section. The author should make efforts to better address the problem and broad impact of MMC (particularly in numbers and facts). Also the logic in introduction has to be reorganized for a more clear story-telling purpose.

Response:  done

2- There are significant flaws in the experiment setup. ZnO-NPs has to be added as an independent group and with MMC to demonstrate that the antigenotoxic effect is solely from ZnO-Algin/NCMs but not Algin or ZnO-NPs alone. Please revise accordingly.

Response

  In the present study we aimed to synthesize and characterize zinc oxide alginate-Nanocomposites (ZnO-Algin/NCMs) and to investigate its ability to reduced MMC induced genotoxicity and so, we didn't investigate the effect of ZnO-NPs alone. But after your notice , we will add these groups in the next research.     

  1. Formatting issues: the font type and size are not the same in some paragraphs of this article, e.g. section 2.4 and section 3.2; figure and table numbers are not properly cited, eg. figure 4; period and comma characters missing in some places; figure title is not informative, e.g. not all images are clearly explained and identified in Figure 5; reference not formatted correctly;

Response:  done

  1. Grammar issues: many sentences in the article are not properly written. Paragraphs are scattered into pieces of information other than being logically and smoothly laid out. Some grammar problems are preventing the reader to understand the content correctly and effortlessly.

Response:  done

Reviewer 2 Report

The methodology adopted in manuscript seems to be adequate, however, it has not been written and presented well. There are several typographical and grammatical mistakes throughout. It is recommended to revise it with regards of these mistakes.

Author Response

Comments and Suggestions for Authors                                                                         

The methodology adopted in manuscript seems to be adequate; however, it has not been written and presented well. There are several typographical and grammatical mistakes throughout. It is recommended to revise it with regards of these mistakes.

Response:  done

Round 2

Reviewer 1 Report

Thanks for making these changes according to the reviewers' comments. Please keep in mind the things that could be improved for future research.